# Historical Review of Studies on Cyrtophorian Ciliates (Ciliophora, Cyrtophoria) from China

**DOI:** 10.3390/microorganisms10071325

**Published:** 2022-06-30

**Authors:** Zhishuai Qu, Hongbo Pan, Jun Gong, Congcong Wang, Sabine Filker, Xiaozhong Hu

**Affiliations:** 1State Key Laboratory of Marine Environmental Science, College of the Environment and Ecology, Xiamen University, Xiamen 361104, China; 2Engineering Research Center of Environmental DNA and Ecological Water Health Assessment, Shanghai Ocean University, Shanghai 201306, China; hbpan@shou.edu.cn; 3School of Marine Sciences, Sun Yat-Sen University, Zhuhai 519082, China; gongj27@mail.sysu.edu.cn; 4Key Laboratory of Mariculture, Ministry of Education, College of Fisheries, Ocean University of China, Qingdao 266003, China; wangcocoh@163.com; 5Department of Molecular Ecology, University of Kaiserslautern, 67663 Kaiserslautern, Germany; filker@rhrk.uni-kl.de

**Keywords:** morphology, phylogeny, SSU rDNA, taxonomy

## Abstract

The subclass Cyrtophoria are a group of morphologically specialized ciliates which mainly inhabit soil, freshwater, brackish water, and marine environments. In this study, we revise more than 50 publications on the taxonomy, phylogeny, and ecology of cyrtophorian ciliates in China since the first publication in 1925, most of which were carried out in coastal areas. The research history can be divided into three periods: the early stage, the Tibet stage, and the molecular stage. To date, 103 morpho-species (147 isolates) have been formally recorded in China, with ciliature patterns described for 82 of them. A species checklist and an illustrated identification key to the genera are provided. A total of 100 small subunit rDNA sequences have been obtained for 74 taxonomic hits (lowest taxonomic rank to species or genus). These sequences are used for the study of molecular phylogeny. Based on these morphological data and molecular phylogeny analyses, we synthesize the understanding of the phylogeny of cyrtophorian ciliates. We hypothesize that the key evolutionary event of cyrtophorian ciliates lies in the separation of the stomatogenesis zone (postoral kineties) from the left kineties, namely, the formation of an independent “sexual organelle”. We, furthermore, briefly summarize the ecological features of cyrtophorian ciliates and provide a comprehensive bibliography of related research from China. Finally, we give an outlook on the future research directions of these taxa.

## 1. Introduction

As evidenced by an increasing number of publications, China is now recognized as one of the hotspots for the study of the taxonomy and systematics of ciliates [1,2,3,4,5]. Among ciliates, cyrtophorians (subclass Cyrtophoria) are a morphologically specialized group that feature a unique combination of morphological characteristics: (1) The cytostome is reinforced by a special buccal structure called the pharyngeal basket which is made up of fibers that are usually organized in the form of nematodesmal rods [6]. (2) The oral ciliature consists of a few short kinetal segments, with usually one preoral kinety (sometimes in several short fragments) and two circumoral kineties (see oral kineties—OK—in Figure 1) [7]. Some species have a degenerated oral structure or more circumoral kineties. During stomatogenesis, the oral kineties are formed by an anticlockwise rotation of the oral anlagen (see stomatogenetic zone—SZ—in Figure 1) [8]. (3) The cilia are mainly restricted to the ventral side, and only a few are located at the anterior part of the dorsal side [7,9]. (4) Some species have organelles adapted to the adhesion on the substrate, such as podites (e.g., *Dysteria* and *Hartmannula*), glandules (e.g., *Trichopodiella*), and finger-like tentacles (e.g., *Lynchella* and *Chlamydonella*) (Figure 1) [7,9,10]. From more “general” cyrtophorians to highly specialized ones, the left kineties tend to degenerate and move to the frontal and right part of the cell. The right kineties are also inclined and mostly restricted to the right margin as a result of the lateral compression of the body in the order Dysteriida (Figure 1).

Most cyrtophorian ciliates live as periphyton and feed on bacteria (including cyanobacteria) and/or eukaryotic microalgae [11,12,13,14]. They are widely distributed in a variety of habitats including soil [15,16], freshwater [11,16,17], marine environments [4,6,9,17,18,19,20,21,22], brackish waters [12,13,23,24,25,26], and even glaciers in Antarctica [27,28]. As active bacterivores, some species commonly occur in eutrophic manmade ecosystems, such as *Chilodonella uncinata*, *Gastronauta* spp., and *Trochilia minuta*, which have been used as bio-indicators for monitoring sludge performance in wastewater treatment plants [11,29,30,31,32]. Several free-living species thrive in mariculture [33,34,35] and freshwater fish aquaculture tanks [36]. A few species even live as obligate parasites on the gills and skin of fish [36,37,38,39,40], or on the mucosa of the blowholes of sea mammals [41,42,43], and hence are target organisms in pathogenic studies [44].

To date, more than 180 cyrtophorian ciliate species are known worldwide, divided into 2 orders, 10 families, and 45 genera [3,8,9]. Several milestones can be identified in the research history: (1) Müller [45] reported two cyrtophorian ciliates (*Trithigmostoma cucullulus* and *Chlamydodon triquetrus*, according to the current nomination). (2) Klein [46] first applied silver impregnation (dry silver nitrate) on ciliates and revealed the ciliature of the cyrtophorian *Chilodonellauncinata*. (3) Kahl [17] summarized more than 80 species with fine descriptions based on previous works (e.g., [47,48,49]) and his own. (4) Deroux [6,18,19,20,21,22] conducted a long-term, exclusive study mainly on cyrtophorian ciliates, documenting about 50 marine species with fine ciliary patterns. (5) Snoeyenbos-West et al. [50] analyzed, for the first time, the systematic position of cyrtophorian ciliates using SSU rDNA sequences.

In this paper, we review the research on cyrtophorian ciliates in China, covering aspects of taxonomy, molecular phylogeny, and ecology, and provide an up-to-date literature guide to related research. We, furthermore, present a revised phylogeny of cyrtophorian ciliates based on existing morphological knowledge and the molecular phylogenetic tree inferred from updated SSU rDNA sequences. As a closing remark, we give an outlook on the future research prospects of this special group of ciliates.

## 2. Morphological Taxonomy

### 2.1. Brief Research History

For this section, we collected and reviewed 50 publications (Table 1) with morphological descriptions, excluding those where the species identification could not be traced and confirmed. To the best of our knowledge, the very first report of cyrtophorian ciliates from China was given by Wang [51] in the year of 1925. He isolated and described three species, *Chilodon caudata* Stokes, 1885, *C. cucullulus* (Müller, 1786) Klein, 1927, and *C. vorax* Stokes, 1887 (current names are *Chilodonella caudata*, *Trithigmostoma cucullulus* and *Chilodonella vorax*, respectively), from freshwater lakes in Nanjing. Only a couple of years later, he amended the list with a new species, *Dysteria amoyensis* Wang, 1934, which was discovered in a marine aquarium in Xiamen [52]. Nie and Ho [53] also established a new species, *Gastronauta fontzoui* Nie and Ho, 1943, extracted from freshwater shrimps; however, they only had a vague morphological description. After an almost three-decade blank, Wang [54] recorded 16 more species, including a new one, during a massive investigation on Mount Everest; unfortunately, only five were described with morphological characteristics. Subsequently, Shen [16] summarized her own novel studies and previous results on ciliate fauna investigation from the Tibet Plateau, and briefly described 22 species (including all the species from Wang [54]). Until then, studies were only conducted using live observation. Since the 2000s, more advanced studies have been based on a combination of live observation and silver staining, mainly focusing on marine and brackish water habitats (e.g., [1,3,4,9,55]). During this period, some scattered studies on soil species and parasites were also published (e.g., [37,38,39,40,56]). Research history in China can, therefore, be divided into three stages (Figure 2): (A) During the early stage from 1925 to 1973, only a handful of studies were published, with six species reported including two new taxa. (B) The Tibet stage (1974–2000) includes the massive faunal investigations summarized by Wang [54] on Mount Everest and the summary by Shen [16] on the Tibet Plateau. In total, they described 22 species, with one new form. This stage also included a few publications by Song and his collaborator [55,57,58]. (C) The molecular stage began after 2000. From then on, most species were described by both live observation and silver staining (mostly protargol staining). During this period, the number of recorded species, new species and species with ciliature patterns increased strongly. The molecular studies also had a rapid growth in this period (see the *Molecular Phylogenetic Studies* section below).

### 2.2. Species List and Classification

As of May 2022, a total of 103 morpho-species (147 isolates) have been reported in China which can be assigned to 8 families and 31 genera (Table 1). Among them, 82 species with ciliature (mainly from protargol staining) have been reported. Three new genera, *Aporthotrochilia* Pan et al., 2012, *Heterohartmannula* Pan et al., 2012, and *Paracyrtophoron* Chen et al., 2012, and thirty-nine new species have been erected. Here, we list the genus classifications regarding the species (with reported morphology) found in China. The class, subclass, order and family assignments are mainly based on Lynn [8] and subsequent modifications (see references in Table 1). Accordingly, Figure 3 and Figure 4 provide an illustrated key to identify the genera existing in China on the basis of morphological characters (mainly ciliary patterns).

A list of cyrtophorian genera recorded in China and their systematic assignments.


Class Phyllopharyngea de Puytorac et al., 1974Subclass Cyrtophoria Fauré-Fremiet in Corliss, 1956
Order Chlamydodontida Deroux, 1976
Family Chilodonellidae Deroux, 1970
*Chilodonella* Strand, 1928*Odontochlamys* Certes, 1891*Pseudochilodonopsis* Foissner, 1979*Phascolodon* Stein, 1859*Trithigmostoma* Jankowski, 1967
Family Chlamydodontidae Stein, 1859
*Chlamydodon* Ehrenberg, 1835*Paracyrtophoron* Chen et al., 2012Family Gastronautidae Deroux, 1994
*Gastronauta* Engelmann in Bütschli, 1889Family Lynchellidae Jankowski, 1968
*Atopochilodon* Kahl, 1933*Chlamydonella* Deroux in Petz et al., 1995*Chlamydonellopsis* Blatterer & Foissner, 1990*Coeloperix* Deroux in Gong & Song, 2004*Lynchella* Jankowski, 1968Family Plesiotrichopidae Deroux, 1976
*Trochochilodon* Deroux, 1976*Incertae sedis**Lophophorina* Penard, 1922Order Dysteriida Deroux, 1976
Family Dysteriidae Claparède & Lachmann, 1859
*Agnathodysteria* Deroux, 1976*Dysteria* Huxley, 1857*Microxysma* Deroux, 1977*Mirodysteria* Kahl, 1933*Trochilia* Dujardin, 1841Family Hartmannulidae Poche, 1913
*Aegyria* Chen et al., 2012*Aporthotrochilia* Pan et al., 2012*Brooklynella* Lom & Nigrelli, 1970*Chlamydonyx* Deroux, 1976*Hartmannula* Poche, 1913*Heterohartmannula* Pan et al., 2012*Orthotrochilia* Deroux in Song, 2003*Trichopodiella* Corliss, 1960*Trochilioides* Chen et al., 2011Family Kyaroikeidae Sniezek & Coats, 1996
*Kyaroikeus* Sniezek, Coats & Small, 1995*Planilamina* Ma et al., 2006




### 2.3. Comments on New Taxa Described from China

Three new genera and thirty-seven new species have been established in China. We carefully examined all these new taxa using the descriptions in the original publications and by checking the deposited specimens, and found that the establishment of one genus was not necessary, and the name of one species was already pre-occupied.

*Paracyrtophoron* Chen et al., 2012, was established mainly because it differed from its closest congener *Cyrtophoron* by “the lack of a fragment near anterior ends of right kineties and transpodial fragments in the posterior portion of the ventral surface” [23]. Here, the “fragment near anterior ends of right kineties” seems to be a structure only described in *Cyrtophoron isagogicum* [6], but not in other *Cyrtophoron* species; thus, it cannot be used as a promising generic difference. Additionally, it is not convincing that the transpodial fragments can be considered as a genus-level discrepancy, because these fragments are not necessarily present among the species of the related genus *Chlamydodon* (e.g., present in *C.*
*pararoseus*, but absent in *C. wilberti*) [26,81]. Therefore, we doubt the necessity and the validation of the establishment of *Paracyrtophoron* from *Cyrtophoron*.

Furthermore, we have some comments on another genus, *Aporthotrochilia* Pan et al., 2012. It has been stated that the difference between *Aporthotrochilia* and *Orthochilia* is that the former has fragments on the right, posterior part of the frontoventral kineties and a higher number of terminal fragments [74]. This description is, however, somewhat unclear. In our opinion, the outer right kineties in *Aporthotrochilia* are interrupted in the middle, which leaves anterior parts (called “several terminal fragments” by the original authors) and posterior parts (corresponding to the posterior fragments). This is supported by L, S and T in the original publication [74], in which the anterior fragments are clearly a part of the outer right kineties. Thus, *Aporthotrochilia* has only one terminal fragment. Nevertheless, this interruption of the outer right kineties could be considered as a sufficient generic difference, and the establishment of the new genus is hence reasonable.

A new species of *Chilodonella*, *C. parauncinata*, was suggested by Qu et al. [59], and the separation from its congeners is undoubtable. However, the authors neglected the preoccupation of the species name, which was also a new species established by Wang [54] from Mount Everest (no deposited type material). This highlights the importance of an extensive literature search, especially for taxonomic study. *C. parauncinata* Wang, 1974, was originally published in Chinese; we provide herein a translation of the vague diagnosis of the poorly known species (no staining information): “Cell length 45–48 µm in vivo; body shape irregularly oval, beak-shaped protrusion to left in anterior end; posterior end more or less rounded, concave on right margin; four right and four or five left kineties; isolated from soil on the Mount Everest”. These two isolates can be clearly distinguished by the numbers of somatic kineties: the form in Qu et al. [59] had five right (vs. four) and six or seven left kineties (vs. four or five). Therefore, *Chilodonella parauncinata* sensu Qu et al., 2015 is a junior primary homonym, and it is permanently invalid according to article 57.2 of the ICZN [84]. Thus, we replace *Chilodonella parauncinata* sensu Qu et al., 2015 with a new name, *Chilodonella apouncinata* nom. nov.***Chilodonella apouncinata* nom. nov.***Chilodonella parauncinata*—Qu et al., J. Eukaryot. Microbiol. 2015, 62, 267–279 (primary homonym, non *Chilodonella parauncinata* Wang, 1974).**ZooBank registration number of present paper.** urn:lsid:zoobank.org:pub:CEB2291F-08EC-4840-8979-01F28FF6DB83.**ZooBank registration number of *Chilodonella apouncinata* nom. nov.** urn:lsid:zoobank.org:act:5E887B0E-B6DC-462A-9C20-9A1C6721CE44.**Type specimen.** See Qu et al. [59] (p. 269, Figure 3H).**Type locality.** See Qu et al. [59] (p. 275).**Deposition of type materials.** See Qu et al. [59] (p. 275).**Etymology.** The species-group name *apouncinata* is a composite of the Greek adjective *apo*- (from) and the species-group name *uncinata*, indicating that the species is similar to *Chilodonella uncinata*.**Morphological description and morphogenesis**. See Qu et al. [59] (pp. 269–271).**Comparison with congeners.** See Qu et al. [59] (pp. 272–273).

## 3. Molecular Phylogenetic Studies

A new perspective on the systematics of cyrtophorid ciliates emerged at the turn of the millennium with the advent of molecular techniques. Molecular research corresponds to the molecular stage of morphological research (stage C in Figure 2). The first two SSU rDNA (small subunit ribosome DNA) sequences from China of the species *Dysteria derouxi* (AY378112) and *Hartmannula derouxi* (AY378113) were released in 2003. Shortly after that, the number of species studied with molecular sequences increased rapidly. By May 2022, 100 SSU rDNA sequences of Chinese origin had been deposited in the GenBank database with to 74 taxonomic hits (lowest taxonomic rank to species or genus), of which 66 had verifiable morphological records (Table 2). The molecular signatures (mainly the SSU rDNA) were then used to study the phylogeny of cyrtophorian ciliates. By this, Li and Song [85] revealed the phylogenetic positions of *Dysteria derouxi* and *Hartmannula derouxi*, the monophyly of families Chilodonellidae, Chlamydodontidae and Dysteriidae, and the clustering of Hartmannulidae with *Isochona* species (subclass Chonotrichia). Gong et al. [70] constructed phylogenetic trees using SSU rDNA and group I introns. Although only a few representatives were involved, the possible close relationship of Hartmannulidae and the subclass Chonotrichia was also indicated.

All subsequent work followed the systematic scheme proposed by Lynn [8] that cyrtophorian ciliates belong to the subclass Cyrtophoria, with two orders assigned: Chlamydodontida Deroux, 1976 and Dysteriida Deroux, 1976. Gao et al. [86] comprehensively analyzed the phylogeny of cyrtophorian ciliates using SSU rDNA sequences from 7 families and 17 genera, and portraited the overall phylogenetic structure for the first time, which then became the template for the following studies: The non-monophyly of the order Chlamydodontida and the monophyly of Dysteriida were confirmed; a new family, Pithitidae Gao et al., 2012, was established based on its special phylogenetic position and unique morphological feature; and the position of the family Plesiotrichopidae was found to be uncertain. However, we agree with Lynn [87] that the sequence of *Plesiotrichopus*, the type genus of Plesiotrichopidae, is still missing; thus, the certainty of the family transfer needs further testing in the future. Following that, Chen et al. [88] conducted complementary analyses with additional SSU rDNA sequences. The phylogenetic topology was consistent with that in Gao et al. [86], but with further discussion on some newly sequenced taxa such as *Brooklynella*. Based on these results, the possible evolutionary pattern of cyrtophorian ciliates was proposed. However, this evolutionary routine was without sufficient references from either morphological data or molecular phylogeny. The authors also showed the prediction of the secondary structures of the hypervariable region 4 (V4) of the SSU rDNA of the representative cyrtophorian genera. Later, Wang et al. [89] analyzed the phylogeny of class Phyllopharyngea—including Cyrtophoria—using two genes: the mitochondrial SSU rDNA (mtSSU rDNA) and the nuclear SSU rDNA. The phylogenetic results were generally consistent with previous works. Recently, Pan et al. [90] employed high-throughput sequencing to obtain the genomic and transcriptomic data of seven species, including five cyrtophorians. The emphasis of this paper, however, was to solve the uncertain phylogenetic position of Synhymenia, and was not focused on cyrtophorian ciliates.

Other works involving molecular phylogeny (SSU rDNA) were combined with morphological descriptions, serving as a complementary method for species identification [12,13,14,24,25,26,37,39,41,59,74,77,78,79,80,81].

In the present work, we reconstruct a comprehensive phylogenetic tree inferred from updated SSU rDNA sequences (Figure 5). Sequence selection and alignment, and phylogenetic tree construction methods are described in the Appendix A. The topology of the present phylogenetic tree is consistent with previous works (e.g., [14,41,59,77,78,79,80,81,85,88,89,90]), and similar conclusions can be drawn: (1) The order Clamydodontida are monophyletic, while the order Dysteriida are non-monophyletic. (2) The subclass Chonotrichia are nested within Dysteriida, on the level of the family Hartmannulidae. (3) Kyaroikeidae display as a subfamily of Dysteriidae. (4) Gastronautidae represent a transitional family among Chilodonellidae, Chlamydodontidae, and Lynchellidae.

## 4. Proposed Phylogeny

With some key members not yet uncovered, as well as missing molecular representatives, a detailed evolutionary relationship (phylogeny) of the subclass Cyrtophoria cannot be drawn satisfactorily. As mentioned above, Chen et al. [88] have attempted to illustrate the evolutionary relationships among the recognized cyrtophorian genera. However, these relationships were not strictly based on morphological and molecular data. For example, the solid separation of the two orders, Chlamydodontida and Dysteriida, were not supported by molecular phylogenetic analyses (Figure 5), and the positions of *Chlamydodon* and *Cyrtophoron* at the end of the Lychellidae branch were confusing. Nevertheless, what we can infer is that there is an obvious discrepancy between the considerable knowledge on the morphological diversity of cyrtophorian ciliates and their phylogeny and evolution. It has been emphasized that the morphogenetic data have to be taken into account in order to reconstruct evolutionary and phylogenetic relationships in ciliates. However, investigations show that the morphogenetic process of cyrtophorians seems quite identical among all groups [6,13,14,19,57,59,91,92], thus providing only limited information. Based on the review of the morphological (including feeding- and motility-associated morphological traits) and phylogenetic works mentioned above, and the one constructed ourselves (Figure 5), we propose the phylogeny of 29 genera from 9 families for which molecular data are available (Figure 6).

The separation of the two orders, Chlamydodontida and Dysteriida, is mainly based on the form of the cell (dorso-ventrally compressed in the former and laterally compressed in the latter) and the posterior adhere organelles (podite and/or glandule absent in the former and present in the latter). However, nothing has been mentioned about the trend of the morphogenesis-related postoral kineties (see stomatogenesis zone—SZ—in Figure 1). In the order Chlamydodontida, the postoral kineties are not well separated from either the left or right kineties (Lynchellidae, Chlamydodontidae, and *Trithigmostoma*), or are merged as the inner part of the left kineties (Chilodonellidae and *Gastronauta*). Despite the highly degenerated left kineties, the postoral kineties are still connected with the left kineties in Hartmannulidae. Then, the postoral kineties tend to separate from the left kineties and become independent in Dysteriidae and Kyaroikeidae. We here hypothesize that the key evolutionary event of cyrtophorian ciliates lies in the separation of the stomatogenesis zone (postoral kineties) from the left kineties, namely, the formation of an independent “sexual organelle”.

As shown in Figure 5, the phylogenetic positions of the families Lynchellidae, Chilodonellidae, Chlamydodontidae, Hartmannulidae, and Dysteriidae are clear, with strong support from abundant morphological and molecular data [86,88,89]. In contrast, the positions of the families Pithitidae, Gastronautidae, Plesiotrichopidae, and Kyaroikeidae appear vague, mainly because of the lack of morphological and/or molecular data. It should be kept in mind that one should not over interpret the topology of phylogenetic trees derived from limited molecular data. Pithitidae and Plesiotrichopidae comprise only a few morpho-species and only one molecular representative each, which largely obscures phylogenetic study. Nevertheless, we agree with Gao et al. [86] that these two families are intermediate groups between the orders Chlamydodontida and Dysteriida, mainly from the aspect of morphological comparison. Similarly, although Gastronautidae has only one sequence positioned within the family Chilodonellidae, this family is valid and represents an intermediate group within the order Chlamydodontida based on its unique combination of morphological characteristics [14]. The separation of Kyaroikeidae (*Kyaroikeus* and *Planilamina*) from Dysteriidae is not well supported by either morphological or molecular data, and Kyaroikeidae species could be seen as highly specialized Dysteriidae adapted to parasitism [41]. Therefore, we place it at the basal position of Dysteriidae.

Several intermediate genera are also recognized: *Trithigmostoma* belongs to the family Chilodonellidae, but shares some characteristics with the Chlamydodontidae family, which is why it is placed at the basal position within the Chilodonellidae [14]. *Brooklynella*, *Trochilioides* and *Microxysma* represent transitional genera between Hartmannulidae and Dysteriidae [86,88] with intellectual morphological characteristics [65,72,75,76].

## 5. Ecology

### 5.1. Sites, Habitats, and Distribution

China is characterized by a vast diversity of habitats, including plateau regions, deserts, rain forests, wetlands, rivers, lakes, coastal areas, and deep sea. Accordingly, it is reasonable to assume that there should also be a high degree of ciliate biodiversity. However, apart from in the very early stage, respective investigations have mostly been carried out along coastal areas (Figure 7), while other habitats are only sporadically studied. The sampled sites can, therefore, be categorized into coastal areas and inland areas. Coastal areas can further be classified into three parts: (1) The northern part (the coasts of the Bohai Sea and the Yellow Sea) is located in a temperate zone. Sampling sites include Qingdao, Yantai, Changyi, and Dongying. (2) The eastern part (the estuary of Yangtze River) includes Shanghai, Ningbo, and Hangzhou. (3) The southern part (the coasts of the Southern China Sea and the estuary of the Pearl River) includes Xiamen, Huizhou, Hong Kong, Shenzhen, Guangzhou, Zhuhai, Zhanjiang, and Haikou. The inland areas cover the Tibet Plateau (including Mount Everest), Gansu province, and cities along the Yangtze River and its tributary (Dali, Liangshan, Luzhou, Hanchuan, Wuhan, Nanjing, and Changzhou).

The investigated areas covered five habitat types or lifestyles (Figure 7), namely, marine, brackish water, freshwater, soil, and parasitism. Studies on the northern coastal areas focused mainly on marine habitats (coastal waters, indoor or open sea mariculture waters), while studies on the eastern and southern coasts were mainly conducted from brackish waters. Freshwater habitats were mainly investigated along the Yangtze River and its tributary, and on the Tibetan Plateau. Most evidence of cyrtophorian parasitic lifestyles also comes from these areas. Soil cyrtophorians were mainly isolated from Gansu and Mount Everest.

Many studies on pelagic and soil habitats indicate that the dispersal and distribution of ciliates follow the moderate endemicity model (e.g., [93]). Apparently, this also holds true for cyrtophorian ciliates. Currently, more than one third of cyrtophorians discovered in China are new and possibly endemic, while the rest can be found in other countries or on other continents as well. The only attempt to analyze the biogeographic distribution patterns of cyrtophorian ciliates was briefly conducted on a well-studied genus: *Chlamydodon* [13]. This work summarized the historical studies on the morphology of this genus worldwide, showed the global distribution of *Chlamydodon* species, and indicated possible cosmopolitan and endemic species. However, as stated by the authors, the analyses were very limited, mainly because of the scattered studies on this genus. This limit also applies to the studies on cyrtophorian ciliates in China. For instance, a cosmopolitan species, *Chlamydodon mnemosyne*, has only been formally reported once in a publication from China [64], but sampled and recorded in master’s and PhD theses several times (by personal communication). This issue prevents a thorough diversity study and also hinders intra- and interspecific comparisons for both morphological and molecular aspects. Therefore, in order to obtain more detailed data (at population level) with a higher geographical resolution, extensive sampling must be carried out in different habitats at a larger scale.

### 5.2. Lifestyles

Most cyrtophorian species display a free-living periphyton life, and can thus be easily sampled by gently scratching the substrate or enrichment with artificial substrates such as glass slides [35,67]. Some species are found mainly in aquaculture water bodies, especially mariculture waters [33,34,35] such as *Dysteria crassipes* [67,79]. A few species exhibit a parasitic lifestyle. *Chilodonella uncinata* [40], *C. piscicola* [37,39], and *C. hexasticha* [38,39,40] have been reported to be opportunistic or obligate parasites on gills and the skin of freshwater fish. Furthermore, two mammal parasites, *Kyaroikeus paracetarius* and *Planilamina ovata*, were isolated from the mucus of an unhealthy captive beluga [41].

### 5.3. Food Source and Feeding Types

Cyrtophorian ciliates mainly feed on bacteria (including cyanobacteria) and/or eukaryotic microalgae, and they are obligate bacterivores, algivores, or omnivores [12,13,14]. This could be inferred by a direct check of food vacuoles or cultivation attempts using grain-enriched bacteria. Some of this information can be used for species identification. For instance, *Pseudochilodonopsis algivora* is filled with algae inclusions, which is a promising character to quickly identify this species [16]. Some species such as *Aegyria oliva*, *A. paroliva*, *Chlamydodon bourlandi*, *C. obliquus*, and *Paracyrtophoron tropicum* have unusual violet digested inclusions that can be used to aid species identification [4,23,26,82]. We have tried to summarize food inclusions by checking descriptions, photomicrographs, and cultivation attempts in the literature. A large part of cyrtophorian ciliates feed on microalgae (e.g., *Chlamydodon*, *Hartmannula*, *Pseudochilonopsis*, and *Trithigmostoma*). This can be best explained by their strong pharyngeal basket which is adapted to undertake microalgae food. Small, spherical algae are mostly observed as food inclusions (e.g., [12,16,55,59]). Large diatoms are also commonly found, and most of them belong to Naviculaceae (e.g., [14,24,80,81]), while *Cyclotella* is casually identified [24]. Thus far, filamentous cyanobacteria have only been reported in *Chlamydodon mnemosyne* [64] and a *Chilodonella* species [59]. Bacterivorous species are also common, especially for those species found in the aquaculture waters (*Chilodonella* spp. and *Dysteria* spp.). These species can be cultivated by the enrichment of bacteria by adding rice or wheat grains. A few species (*Pseudochilodonopsis marina*, *Dysteria brasiliensis*, and *Gastronauta paraloisi*) have been reported to feed on both bacteria and small, spherical algae, and thus are possibly omnivorous [14,55,67,68,77]. No predation has been found in cyrtophorian ciliates. Although *Aegyria oliva* has been reported once to contain small scuticociliates and *Aspidisca* in its inclusions [4], this was likely the result of non-selective feeding, and no predatory behavior was confirmed.

### 5.4. Abundance

Cyrtophorian ciliates have relatively low abundance in natural habitats compared to other ciliates, but some species (bacterivores or omnivores) are found to be quite abundant in aquaculture or saprobic environments, such as *Dysteria* spp. [26,67,77] and *Gastronauta paraloisi* [14]. Although some diatom consumers, such as *Trithigmostoma*, may be dominant when food diatoms are enriched [14,81], the population density of obligate algivores is usually low.

### 5.5. Others

A few studies have reported that dysteriid species (mainly *Dysteria*) have possible ectosymbiotic bacteria (most likely bacterial epibionts) on cell surface, e.g., *Aegyria foissneri* [25], *Dysteria brasiliensis* [67], *D. compressa* [80], *D. crassipes* [26,67,79], *D. lanceolata* [72], *D. monostyla* [26], *D. paraprocera* [79], and *D. subtropica* [68,79]. The occurrence and the type of these bacteria seem to be environment-induced [67,77]. However, the exact function of these bacteria is yet unknown.

## 6. Prospects

With almost a century of research history, studies on the taxonomy, phylogeny, and ecology of cyrtophorian ciliates in China have accumulated substantial results. In addition to this, we offer an outlook on the future research of this group of ciliates.

Despite the considerable number of studies published in China, as described above, it is clear that there are still many unknown species that prevent a more detailed and cohesive systematic scheme of Cyrtophoria. Thus, more investigations into fauna need to be conducted on a large regional scale with different habitats. In contrast to massive investigations from marine and brackish water habitats, sampling from freshwater and soil as well as extreme environments such as hypersaline lakes (mostly on the Tibetan Plateau, western China) and cold regions (northern China) are urgently needed;The de facto standard for the taxonomic study of ciliates combining morphology (live observation and silver staining) and molecular phylogeny [94] has been well practiced for Cyrtophoria over the last decade. As high-throughput sequencing becomes cost-efficient and more effective bioinformatic tools are developed, it is possible to use these techniques to perform phylogenomic reconstruction, as indicated by [91]. Thus, in the near future, genomic/transcriptomic data should also be included in taxonomy or phylogenetic study routines to achieve higher phylogenetic resolution;Similarly to other ciliate groups, species separation/circumscription is still a major problem for the taxonomy of cyrtophorian ciliates. Different geographical populations of the same species should be recorded, described and compared on the aspects of morphology and different marker genes;Attention should be paid to the ectosymbiotic bacteria (or bacterial epibionts) on the cell surface of *Dysteria*, with emphasis on trophic function as well as possible phylogenetic signals to their hosts;Detailed ecological roles (niches) of cyrtophorian ciliates should be elucidated. This could be performed by annual or seasonal sampling, cultivation experiments, and checking food vacuoles or fluorescence in situ hybridization (FISH) to detect species occurrence, food inclusion, and trophic relationships (autecology).

## Figures and Tables

**Figure 1 microorganisms-10-01325-f001:**
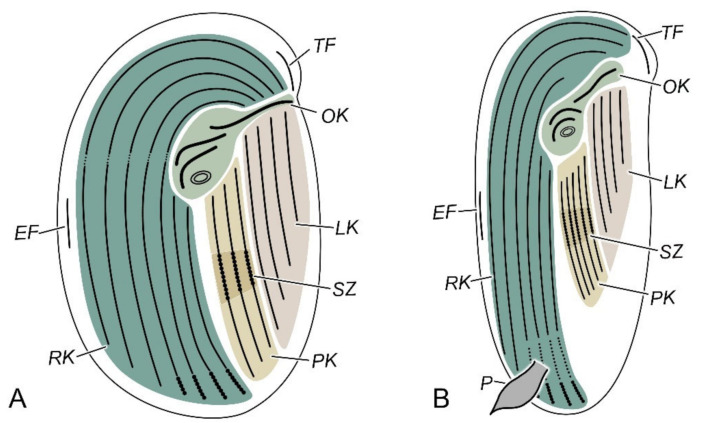
Schematic ciliary patterns of cyrtophorian ciliates. (**A**) Order Chlamydodontida. (**B**) Order Dysteriida. Abbreviations: EF—equatorial fragment; LK—left kineties; OK—oral kineties; P—podite; PK—postoral kineties; RK—right kineties; SZ—stomatogenetic zone; and TF—terminal fragment.

**Figure 2 microorganisms-10-01325-f002:**
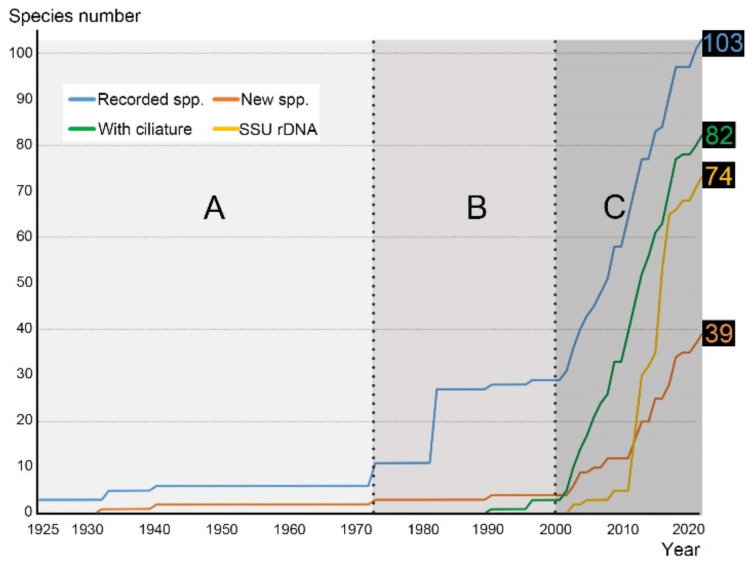
Timeline and accumulated numbers of studied cyrtophorian ciliates (species) in China. The four lines represent the numbers of recorded species, new species, species with ciliature information, and taxonomic hits (lowest taxonomic rank to species or genus) with SSU rDNA sequences. A, B, and C represent three study periods: the early stage (**A**) 1925–1973)), the Tibet stage (**B**) 1974–2000)), and the molecular stage (**C**) since the 2000s)).

**Figure 3 microorganisms-10-01325-f003:**
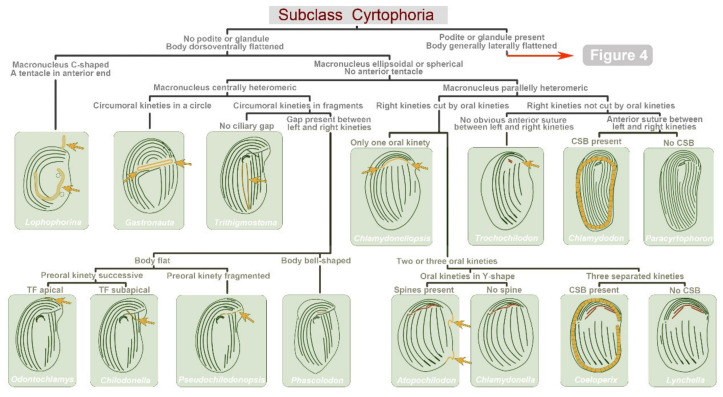
Illustrated key for the identification to cyrtophorian genera found in China. All the illustrations are original. The arrows and colored circles indicate dichotomic characteristics. Abbreviations: CSB—cross-striated band; TF—terminal fragment.

**Figure 4 microorganisms-10-01325-f004:**
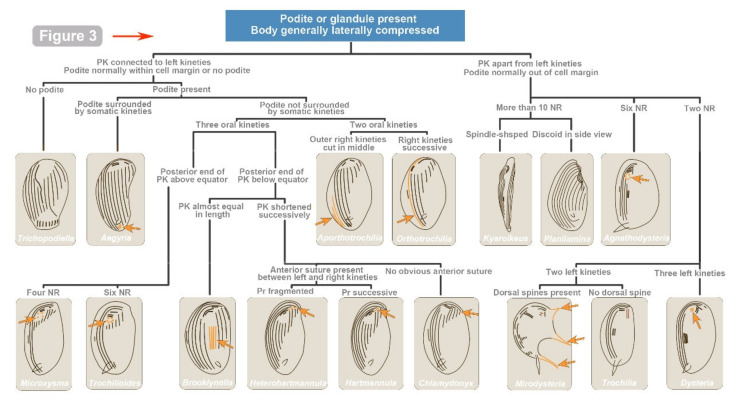
Illustrated key for the identification to cyrtophorian genera found in China (Figure 3 continued). All the illustrations are original. The arrows and colored structures indicate dichotomic characteristics. Abbreviations: NR—nematodesmal rods; PK—postoral kineties; and Pr—preoral kinety.

**Figure 5 microorganisms-10-01325-f005:**
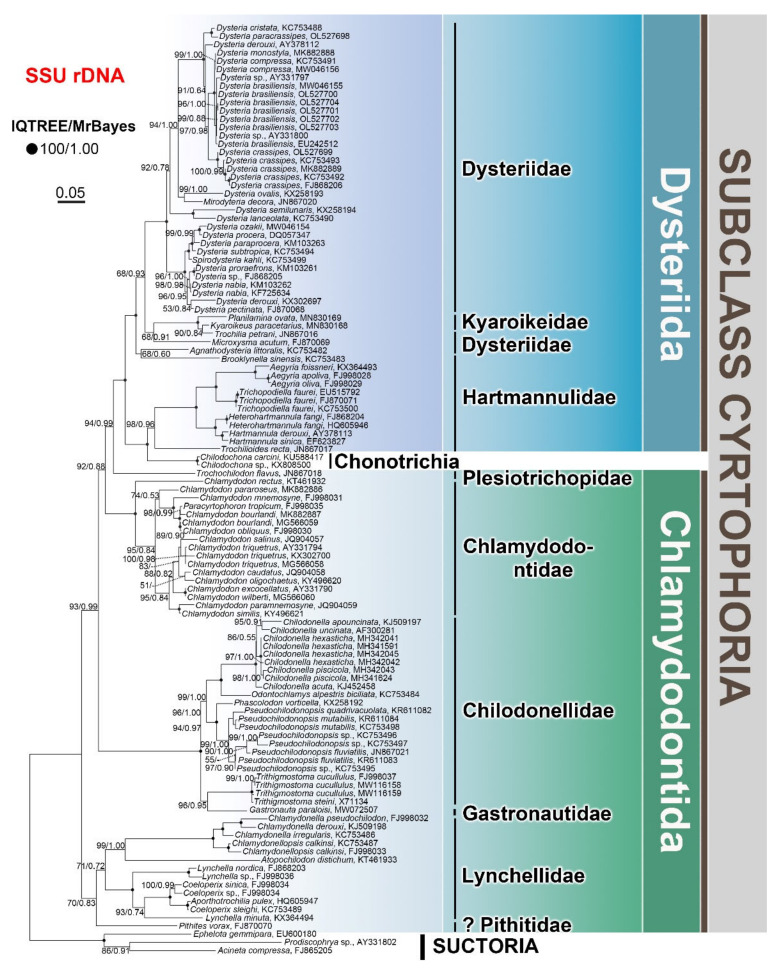
Phylogenetic tree inferred from SSU rDNA sequences. The trees were reconstructed by two algorithms: IQTREE and MrBayes. Support values from the two methods are provided at the branching points (IQTREE/MrBayes). Bold dots at the branching points indicate full support from both analyses. “-” indicates discrepancy between the topologies of IQTREE and the MrBayes trees. The scale bar represents five substitutions per hundred nucleotide positions.

**Figure 6 microorganisms-10-01325-f006:**
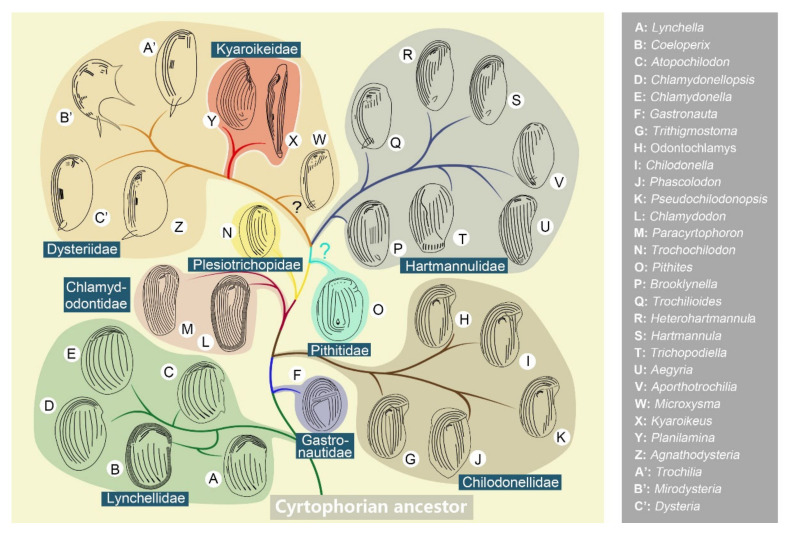
Evolutionary pattern of the subclass Cyrtophoria inferred from species with both morphological and molecular data. All illustrations are original.

**Figure 7 microorganisms-10-01325-f007:**
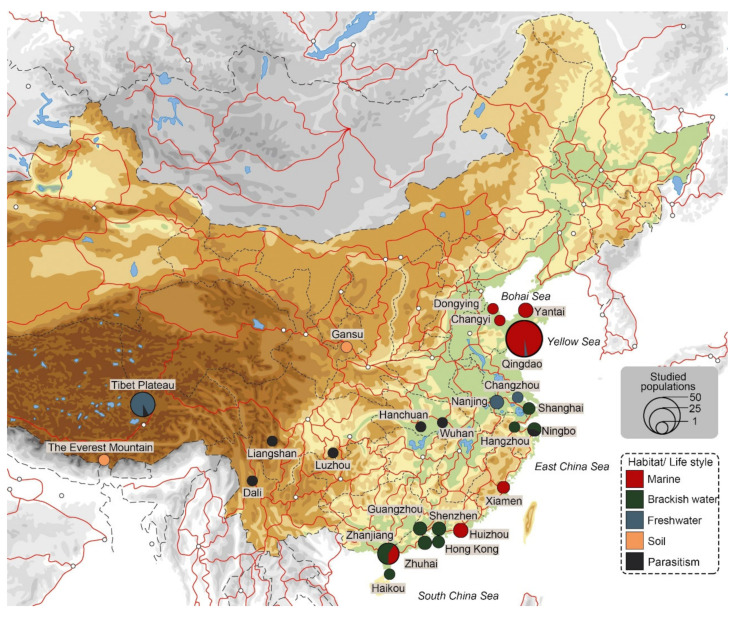
Sampling sites and habitat categories (or lifestyles) of studied cyrtophorian ciliate populations from China. The dots represent study sites, and the sizes of the dots indicate the number of populations studied. Colors represent different habitat types or lifestyles.

**Table 1 microorganisms-10-01325-t001:** Checklist of cyrtophorian ciliate species/isolates with morphological descriptions from China. New species are highlighted in bold. Species are presented with current names. * Ref. [4] by Song et al. summarizes 32 species found by Song’s group (Ocean University of China) from 1991 to 2008. Here, only seven species/populations not published before 2009 are listed. † This species was isolated from marine water, which is different from those populations of *Chilodonella uncinata* isolated from soil or freshwater habitats; thus, we count it as a different species. § Previous name was *Chilodonella parauncinata* [59].

Publication	Species Name (Current Name)	Ciliature	Site	Habitat
Wang (1925) [51]	*Chilodonella caudata* Stokes, 1885	-	Nanjing	Freshwater
	*Chilodonella vorax* Stokes, 1887	-	Nanjing	Freshwater
	*Trithigmostoma cucullulus* (Müller, 1786) Jankowski, 1967	-	Nanjing	Freshwater
Wang (1934) [52]	***Dysteria amoyensis* Wang, 1934**	-	Xiamen	Marine
	*Hartmannula entzi* Kahl, 1931	-	Xiamen	Marine
Nie and Ho (1943) [53]	***Gastronauta fontzoui* Nie & Ho, 1943**	-	-	Freshwater shrimps
Wang (1974) [54]	*Chilodonella aplanata* Kahl, 1931	-	Mount Everest	Soil
	***Chilodonella parauncinata* Wang, 1974**	-	Mount Everest	Soil
	*Chilodonella uncinata* (Ehrenberg, 1838) Strand, 1928	-	Mount Everest	Soil
	*Odontochlamys convexa* (Kahl, 1931) Blatterer & Foissner, 1992	-	Mount Everest	Soil
	*Pseudochilodonopsis algivora* (Kahl, 1931) Foissner, 1979	-	Mount Everest	Soil
Shen (1983) [16]	*Chilodonella aplanata* Kahl, 1931	-	Tibetan plateau	Freshwater
	*Trithigmostoma bavariensis* (Kahl, 1931) Foissner, 1987	-	Tibetan plateau	Sodium sulfate lake
	*Chilodonella capucina* (Penard, 1922) Kahl, 1931	-	Tibetan plateau	Freshwater
	*Chilodonella dentata* Fauguè, 1876	-	Tibetan plateau	Freshwater
	*Chilodonella fluviatilis* Stokes, 1885	-	Tibetan plateau	Freshwater
	*Chilodonella granulate* Penard, 1922	-	Tibetan plateau	Freshwater
	*Chilodonella nana* Kahl, 1928	-	Tibetan plateau	Freshwater
	*Chilodonella parauncinata* Wang, 1974	-	Tibetan plateau	Freshwater
	*Chilodonella piscicola* Zacharias, 1894	-	Tibetan plateau	Freshwater
	*Chilodonella turgidula* Penard, 1922	-	Tibetan plateau	Freshwater
	*Chilodonella uncinata* (Ehrenberg, 1838) Strand, 1928	-	Tibetan plateau	-
	*Chlamydonellopsis calkinsi* (Kahl, 1928) Blatterer & Foissner, 1990	-	Tibetan plateau	Freshwater
	*Lophophorina capronata* Penard, 1922	-	Tibetan plateau	Freshwater
	*Odontochlamys convexa* (Kahl, 1931) Blatterer & Foissner, 1992	-	Tibetan plateau	Freshwater
	*Odontochlamys gourandi* Certes, 1891	-	Tibetan plateau	-
	*Phascolodon vorticella* Stein, 1859	-	Tibetan plateau	Freshwater
	*Pseudochilodonopsis algivora* (Kahl, 1931) Foissner, 1979	-	Tibetan plateau	Freshwater
	*Pseudochilodonopsis labiata* (Stokes, 1891) Packroff, 1988	-	Tibetan plateau	Freshwater
	*Trithigmostoma cucullulus* (Müller, 1786) Jankowski, 1967	-	Tibetan plateau	Freshwater
	*Trochilia minuta* (Roux, 1901) Kahl, 1931	-	Tibetan plateau	Freshwater
	*Trochilia palustris* Stein, 1859	-	Tibetan plateau	Freshwater
	*Trochilia sulcata* (Claparède & Lachmann, 1858)	-	Tibetan plateau	Freshwater
Song (1991) [55]	***Pseudochilodonopsis marina* Song, 1991**	√	Qingdao	Marine
Song (1997) [57]	*Chilodonella uncinata* (Ehrenberg, 1838) Strand, 1928	√	Qingdao	Freshwater
Song and Packroff (1997) [58]	*Dysteria brasiliensis* Faria et al., 1922	√	Qingdao	Marine
Gong et al. (2002) [60]	*Dysteria cristata* (Gourret & Roeser, 1888) Kahl, 1931	√	Zhanjiang	Marine
	*Dysteria monostyla* (Ehrenberg, 1838) Kahl, 1931	√	Qingdao	Marine
Song (2003) [61]	***Chlamydonella derouxi* Song, 2003**	√	Qingdao	Marine
	*Orthotrochilia piluta* (Deroux, 1976) Song, 2003	√	Qingdao	Marine
Gong and Song (2003) [33]	***Dysteria magna* Gong & Song, 2003**	√	Qingdao	Mariculture water
	*Dysteria procera* Kahl, 1931	√	Qingdao	Marine
Gong et al. (2003) [35]	*Dysteria pusilla* (Claparède & Lachmann, 1859) Kahl, 1931	√	Qingdao	Mariculture water
Gong and Song (2004) [34]	***Coeloperix sleighi* Gong & Song, 2004**	√	Qingdao	Mariculture water
Gong and Song (2004) [62]	*Hartmannula angustipilosa* Deroux & Dragesco, 1968	√	Qingdao	Mariculture water
	***Hartmannula derouxi* Gong & Song, 2004**	√	Qingdao	Mariculture water
Gong and Song (2004) [63]	*Dysteria derouxi* Gong & Song, 2004	√	Qingdao	Mariculture water
Gong et al. (2005) [64]	*Chlamydodon mnemosyne* Ehrenberg, 1835	√	Qingdao	Mariculture water
	*Chlamydodon obliquus* Kahl, 1931	√	Qingdao	Marine
	*Chlamydodon triquetrus* (Müller, 1786) Kahl, 1931	√	Qingdao	Mariculture water
Gong and Song (2006) [65]	***Brooklynella sinensis* Gong & Song, 2006**	√	Qingdao	Mariculture water
Gong and Song (2006) [66]	*Chlamydonella derouxi* Song, 2003	√	Qingdao	Mariculture water
	*Chlamydonella pseudochilodon* (Deroux, 1970) Petz et al., 1995	√	Qingdao	Mariculture water
	*Chlamydonellopsis calkinsi* (Kahl, 1928) Blatterer & Foissner, 1990	√	Qingdao	Mariculture water
Gong et al. (2007) [67]	*Dysteria brasiliensis* Faria et al., 1922	√	Qingdao	Marine
	*Dysteria crassipes* Claparède & Lachmann, 1859	√	Qingdao	Mariculture water
	*Dysteria pectinata* (Nowlin, 1913) Kahl, 1931	√	Qingdao	Marine
	*Dysteria semilunaris* (Gourret & Roeser, 1886) Kahl, 1931	√	Qingdao	Mariculture water
Liu et al. (2008) [68]	*Dysteria subtropica* Qu et al., 2015	√	Qingdao	Marine
	*Pseudochilodonopsis marina* Song, 1991	√	Huizhou	Marine
Shao et al. (2008) [69]	***Hartmannula sinica* Shao et al., 2008**	√	Qingdao	Mariculture water
Gong et al. (2008) [70]	***Trichopodiella faurei* Gong et al., 2008**	√	Qingdao	Marine
Fan et al. (2009) [71]	*Chlamydodon obliquus* Kahl, 1931	√	Huizhou	Marine
	*Dysteria derouxi* Gong & Song, 2004	√	Huizhou	Marine
* Song et al. (2009) [4]	*Aegyria oliva* Claparede & Lachmann, 1859	√	Qingdao	Marine and Mariculture
	*Agnathodysteria littoralis* Deroux, 1976	√	Qingdao	Mariculture water
	† *Chilodonella* cf. *uncinata* (Ehrenberg, 1838) Strand, 1928	√	Qingdao	Marine
	*Microxysma acutum* Deroux, 1976	√	Qingdao	Marine
	*Orthotrochilia agamalievi* (Deroux, 1976) Song, 2003	√	Qingdao	Marine
	*Trochilia petrani* Dragesco, 1966	√	Qingdao	Marine
	*Trochilia sigmoides* Dujardin, 1841	√	Qingdao	Mariculture water
Ning et al. (2009) [56]	*Pseudochilodonopsis quadrivacuolata* Qu et al., 2015	√	Gansu	Soil
Chen et al. (2011) [72]	*Chlamydonyx paucidentatus* Deroux, 1976	√	Qingdao	Mariculture water
	*Dysteria lanceolata* Claparède & Lachmann, 1859	√	Qingdao	Mariculture water
	*Lynchella nordica* Jankowski, 1968	√	Qingdao	Mariculture water
Pan et al. (2011) [73]	*Dysteria derouxi* Gong & Song, 2004	√	Qingdao	Marine
	*Dysteria legumen* (Dujardin, 1841) Kahl, 1931	√	Qingdao	Marine
	*Dysteria proraefrons* Clark, 1865	√	Qingdao	Marine
	*Mirodysteria decora* Deroux, 1976	√	Qingdao	Marine
Chen et al. (2012) [23]	** *Aegyria rostellum* ** **Chen et al., 2012**	√	Shenzhen	Brackish water
	***Paracyrtophoron tropicum* Chen et al., 2012**	√	Shenzhen	Brackish water
Hu (2012) [38]	*Chilodonella hexasticha* Kiernik, 1909	√	Luzhou	Ectoparasite of fish
Pan et al. (2012) [74]	*Aporthotrochilia pulex* (Deroux, 1976) Pan et al., 2012	√	Zhanjiang	Marine
	***Heterohartmannula fangi* Pan et al., 2012**	√	Zhanjiang	Marine
	***Trochilia alveolata* Pan et al., 2012**	√	Hong Kong	Brackish water
	*Trochochilodon flavus* Deroux, 1976	√	Qingdao	Marine
Pan et al. (2013) [12]	***Chlamydodon caudatus* Pan et al., 2013**	√	Guangzhou	Brackish water
	***Chlamydodon paramnemosyne* Pan et al., 2013**	√	Zhanjiang	Mariculture water
	***Chlamydodon salinus* Pan et al., 2013**	√	Changyi	Mariculture water
Pan et al. (2013) [75]	***Orthotrochilia sinica* Pan et al., 2013**	√	Qingdao	Marine
	*Trochilioides recta* (Kahl, 1923) Chen et al. 2011	√	Qingdao	Marine
	*Trochilioides tenuis* (Deroux, 1976) Chen et al., 2011	√	Qingdao	Marine
Zhao et al. (2014) [76]	*Coeloperix sleighi* Gong & Song, 2004	√	Zhanjiang	Marine
Deng et al. (2015) [37]	*Chilodonella piscicola* Zacharias, 1894	√	Tibet	Ectoparasite of fish
Qu et al. (2015) [77]	*Pseudochilodonopsis fluviatilis* Foissner, 1988	√	Yantai	Mariculture water
	*Pseudochilodonopsis mutabilis* Foissner, 1981	√	Zhanjiang	Brackish water
	***Pseudochilodonopsis quadrivacuolata* Qu et al., 2015**	√	Zhuhai	Brackish water
Qu et al. (2015) [59]	**§ *Chilodonella apouncinata* nom. nov.**	√	Qingdao	Freshwater
	*Chlamydonella derouxi* Song, 2003	√	Qingdao	Marine
	*Chlamydonella derouxi* Song, 2003	√	Zhanjiang	Brackish water
	*Chlamydonella irregularis* Qu et al., 2015	√	Qingdao	Mariculture water
Qu et al. (2015) [78]	*Dysteria brasiliensis* Faria et al., 1922	√	Yantai	Mariculture water
	*Dysteria brasiliensis* Faria et al., 1922	√	Zhanjiang	Brackish water
	*Dysteria crassipes* Claparède & Lachmann, 1859	√	Dongying	Mariculture water
	*Dysteria cristata* (Gourret & Roeser, 1888) Kahl, 1931	√	Zhuhai	Brackish water
	*Dysteria derouxi* Gong & Song, 2004	√	Qingdao	Marine
	*Dysteria nabia* Park & Min, 2014	√	Zhanjiang	Brackish water
	***Dysteria paraprocera* Qu et al., 2015**	√	Zhanjiang	Brackish water
	*Dysteria proraefrons* Clark, 1865	√	Yantai	Mariculture water
	***Dysteria subtropica* Qu et al., 2015**	√	Huizhou	Marine
Pan et al. (2016) [24]	*Chlamydonellopsis calkinsi* (Kahl, 1928) Blatterer & Foissner, 1990	√	Shanghai	Brackish water
	*Dysteria ovalis* (Gourret & Roeser, 1886) Kahl, 1931	√	Zhejiang	Marine
	*Dysteria semilunaris* (Gourret & Roeser, 1888) Kahl, 1931	√	Shanghai	Brackish water
	*Phascolodon vorticella* Stein, 1859	√	Changzhou	Freshwater
Pan et al. (2017) [79]	*Atopochilodon distichum* Deroux, 1976	√	Hong Kong	Brackish water
	*Chlamydodon rectus* Ozaki & Yagiu, 1941	√	Hangzhou	Brackish water
	***Coeloperix sinica* Pan et al., 2017**	√	Qingdao	Marine
	*Dysteria compressa* (Gourret & Roeser, 1886) Kahl, 1931	√	Zhanjiang	Marine/brackish water
	*Odontochlamys alpestris biciliata* Foissner et al., 2002	√	Guangzhou	Brackish water
Qu et al. (2017) [25]	***Aegyria foissneri* Qu et al., 2017**	√	Zhanjiang	Brackish water
	***Lynchella minuta* Qu et al., 2017**	√	Zhuhai	Brackish water
Chen et al. (2018) [80]	***Aegyria apoliva* Chen et al., 2018**	√	Qingdao	Marine
	*Trithigmostoma cucullulus* (Müller, 1786) Jankowski, 1967	√	Guangzhou	Brackish water
Li et al. (2018) [39]	*Chilodonella hexasticha* Kiernik, 1909	√	Wuhan; Dali; Jiangsu; Hanchuan; Liangshan	Ectoparasite of fish
	*Chilodonella piscicola* Zacharias, 1894	√	Wuhan; Dali; Jiangsu; Hanchuan; Liangshan	Ectoparasite of fish
Qu et al. (2018) [13]	***Chlamydodon crassidens* Qu et al., 2018**	√	Qingdao	Marine
	***Chlamydodon oligochaetus* Qu et al., 2018**	√	Qingdao	Marine
	***Chlamydodon similis* Qu et al., 2018**	√	Shenzhen	Brackish water
Qu et al. (2018) [81]	***Chlamydodon bourlandi* Qu et al., 2018**	√	Zhanjiang	Brackish water
	*Chlamydodon triquetrus* (Müller, 1786) Kahl, 1931	√	Yantai	Mariculture water
	***Chlamydodon wilberti* Qu et al., 2018**	√	Zhanjiang	Brackish water
Wang et al. (2019) [26]	*Chlamydodon bourlandi* Qu et al., 2018	√	Qingdao	Marine
	***Chlamydodon pararoseus* Wang et al**., **2019**	√	Qingdao	Marine
	*Dysteria crassipes* Claparède & Lachmann, 1859	√	Zhanjiang	Brackish water
	*Dysteria monostyla* (Ehrenberg, 1838) Kahl, 1931	√	Haikou	Marine/brackish water
Wang et al. (2019) [40]	*Chilodonella uncinata* (Ehrenberg, 1838) Strand, 1928	√	Hubei	Ectoparasite of fish
	*Chilodonella hexasticha* Kiernik, 1909	√	Hubei	Ectoparasite of fish
Jin et al. (2021) [41]	***Kyaroikeus paracetarius* Jin et al., 2021**	√	Ningbo	Parasite of beluga whale
	*Planilamina ovata* Ma et al., 2006	√	Ningbo	Parasite of beluga whale
Qu et al. (2021) [14]	***Gastronauta paraloisi* Qu et al., 2021**	√	Shenzhen	Freshwater
	*Trithigmostoma cucullulus* (Müller, 1786) Jankowski, 1967	√	Shenzhen, Qingdao, Zhanjiang	Freshwater
Wang et al. (2021) [82]	***Dysteria brasiliensis* Faria et al., 1922**	√	Haikou	Brackish water
	*Dysteria compressa* (Gourret & Roeser, 1886) Kahl, 1931	√	Haikou	Brackish water
	*Dysteria ozakii* Wang et al., 2021	√	Qingdao	Marine
Zhao et al. (2022) [83]	*Dysteria brasiliensis* Faria et al., 1922	√	Ningbo	Brackish water
	*Dysteria crassipes* Claparède & Lachmann, 1859	√	Ningbo	Brackish water
	***Dysteria paracrassipes* Zhao et al., 2022**	√	Ningbo	Brackish water
**Sum**	103 species, 147 populations, 39 new species	82	-	-

**Table 2 microorganisms-10-01325-t002:** NCBI-deposited SSU rDNA sequences of cyrtophorian ciliates from China. One hundred sequences in total belonging to seventy-four cyrtophorian taxonomic hits.

Taxonomic Hit	Accession(s)	Taxonomic Hit	Accession(s)
*Aegyria apoliva*	FJ998028	*Dysteria ovalis*	KX258193
*Aegyria foissneri*	KX364493	*Dysteria ozakii*	MW046154
*Aegyria oliva*	FJ998029	*Dysteria paracrassipes*	OL527698
*Agnathodysteria littoralis*	KC753482	*Dysteria paraprocera*	KM103263
*Aporthotrochilia pulex*	HQ605947	*Dysteria pectinata*	FJ870068
*Atopochilodon distichum*	KT461933	*Dysteria procera*	DQ057347
*Brooklynella sinensis*	KC753483	*Dysteria proraefrons*	KM103261
§ *Chilodonella apouncinata*	KJ509197	** Dysteria* sp.	FJ868205
*Chilodonella hexasticha*	MH342041, MH342042, MH342045, MH341591	*Dysteria semilunaris*	KX258194
*Chilodonella piscicola*	MH341624, MH342043	*Dysteria subtropica*	KC753494
*Chlamydodon bourlandi*	MG566059, MK882887	*Gastronauta paraloisi*	MW072507
*Chlamydodon caudatus*	JQ904058	*Hartmannula derouxi*	AY378113
*Chlamydodon mnemosyne*	FJ998031	*Hartmannula sinica*	EF623827
*Chlamydodon obliquus*	FJ998030	*Heterohartmannula fangi*	HQ605946, FJ868204
*Chlamydodon oligochaetus*	KY496620	*Kyaroikeus paracetarius*	MN830168
*Chlamydodon paramnemosyne*	JQ904059	*Lynchella minuta*	KX364494
*Chlamydodon pararoseus*	MK882886	*Lynchella nordica*	FJ998036
*Chlamydodon rectus*	KT461932	** Lynchella*_sp.	FJ998036
*Chlamydodon salinus*	JQ904057	*Microxysma acutum*	FJ870069
*Chlamydodon similis*	KY496621	*Mirodysteria decora*	JN867020
*Chlamydodon triquetrus*	KX302700, MG566058	*Odontochlamys alpestris biciliata*	KC753484
*Chlamydodon wilberti*	MG566060	*Paracyrtophoron tropicum*	FJ998035
*Chlamydonella derouxi*	KJ509198	*Phascolodon vorticella*	KX258192
*Chlamydonella irregularis*	KC753486	** Pithites vorax*	FJ870070
*Chlamydonella pseudochilodon*	FJ998032	*Planilamina ovata*	MN830169
*Chlamydonellopsis calkinsi*	FJ998033, KC753487	*Pseudochilodonopsis fluviatilis*	JN867021, KR611083
*Coeloperix sinica*	FJ998034	*Pseudochilodonopsis mutabilis*	KR611084, KC753498
*Coeloperix sleighi*	KC753489	*Pseudochilodonopsis quadrivacuolata*	KR611082
** Coeloperix* sp.	FJ998034	** Pseudochilodonopsis* sp.1	KC753495
*Dysteria brasiliensis*	EU242512 MW046155, OL527700-OL5277004	** Pseudochilodonopsis* sp.2	KC753496
*Dysteria compressa*	KC753491, MW046156	** Pseudochilodonopsis* sp.3	KC753497
*Dysteria crassipes*	FJ868206, KC753492, KC753493, MK882889, OL527699	** Spirodysteria kahli*	KC753499
*Dysteria cristata*	KC753488	*Trichopodiella faurei*	EU515792, KC753500, FJ870071
*Dysteria derouxi*	AY378112, KX302697	*Trithigmostoma cucullulus*	FJ998037, MW116158, MW116159
*Dysteria lanceolata*	KC753490	*Trochilia petrani*	JN867016
*Dysteria monostyla*	MK882888	*Trochilioides recta*	JN867017
*Dysteria nabia*	KM103262	*Trochochilodon flavus*	JN867018

* No corresponding morphological data. § The deposited species name in NCBI is *Chilodonella parauncinata*.

## Data Availability

SSU rDNA were obtained from the NCBI database, the accession numbers are shown in Figure 5.

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
