# Peer review of "Historical Review of Studies on Cyrtophorian Ciliates (Ciliophora, Cyrtophoria) from China"

_microorganisms, 2022, doi:10.3390/microorganisms10071325_

Round 1

Reviewer 1 Report

This is an excellent contribution to the field of ciliates systematics. An important goup of cliates, Cirtophorians, are summarised and reviewed in the manuscript. The paper is well structured and clear written. I have no negative feellings on this paper and suggest to accept it for the publication in Microorganisms 

Author Response

Thank you so much for your time and effort!

Reviewer 2 Report

The paper need minor modifications.

Title: can you find something more captivating?

Abstract: 

- line 25: I suggest to substitute the term "demonstrated" with "propose" or "hypothesize", or something similar.

Introduction: being a review, I suggest to include a table that reports a descriptin of terms as "nematodesmal rods" and "kinetal segments" and other very specialist definition.

Figure 1: PK is not described

Paragraph 2.3. Comments on New Taxa Described from China: again, I suggest a more captivating title. 

the same for the chapter: 5. Proposed Phylogeny

Figure 3: what the arrows and the yellow circle indicate? Please, report also in the fig. legend.

Figure 4:  what the arrows indicate? Please, report also in the fig. legend.

Chapter: 5. Proposed Phylogeny, again, I suggest a more captivating title. Furthermore, how the proposed phylogeny is related with that from E.Gentekaki et al (2014)?

7. Prospect. What does it means, Perspectives?

In general, all titles need a review.

Author Response

Reviewer 2

The paper need minor modifications.

Title: can you find something more captivating?

Authors: We changed the title to “Historical Review of Studies on Cyrtophorian Ciliates (Ciliophora, Cyrtophoria) from China”.

Abstract: 

- line 25: I suggest to substitute the term "demonstrated" with "propose" or "hypothesize", or something similar.

Authors: Thanks! We have changed the term to “hypothesize” in R1 (line 31).

Introduction: being a review, I suggest to include a table that reports a descriptin of terms as "nematodesmal rods" and "kinetal segments" and other very specialist definition.

Authors: Thanks for the suggestion. Actually, we already gave a glossary in the end of the manuscript explaining relative terminology as well as their abbreviations (lines 851–887).

Figure 1: PK is not described

Author: Thanks for the observation. Now all the abbreviations are explained (lines 63–65).

Paragraph 2.3. Comments on New Taxa Described from China: again, I suggest a more captivating title. The same for the chapter: 5. Proposed Phylogeny

Authors: Many thanks for your suggestions. After consideration, we decided to keep our titles in brief, which will be easier for the readers to locate interested content.

Figure 3: what the arrows and the yellow circle indicate? Please, report also in the fig. legend.

Figure 4:  what the arrows indicate? Please, report also in the fig. legend.

Authors: The arrows and colored circles indicate the dichotomic characteristics. We have added the indication in the legends of Figures 3 and 4 (lines 231, 235, 236). Thanks!

Chapter: 5. Proposed Phylogeny, again, I suggest a more captivating title. Furthermore, how the proposed phylogeny is related with that from E.Gentekaki et al (2014)?

Authors: Thanks again for the suggestion. As explained above, we decided to keep the simple title as “Propesed Phylogeny”. As we understand, the paper by Gentekaki et al. (2014) is dealing with the phylogenetic positions of 12 ciliates with emphasis of Protocruzia, and no cyrtophorian ciliates were involved.

  1. Prospect. What does it means, Perspectives?

Authors: Here, the “Prospect” means outlook. Based on our review, we list some future research interests in this field.

In general, all titles need a review.

Authors: Thanks again for aa the comments and suggestions which greatly improved our manuscript.

Reviewer 3 Report

This manuscript is a review of the publications on the cyrtophorian ciliates from China. Certainly, the studies on ciliates have experienced an exponential increase in the last two decades in China, and this kind of reviews, with the schematic illustrations and glossary of the terminology, are also useful for the formation of researchers. However, this review has an important deficiency. There is no a phylogenetic tree (SSU rRNA gene) of this group of ciliates. This is needed because the authors proposed a classification of the cyrtophorian ciliates, and also scheme of the evolution. The reader needs to see an updated phylogenetic tree where to verify the coherence between the classification, evolution, and molecular data. This is the main reason to request a major revision. After that, I will recommend the publication of this review.

Minor comments:

line 17: Abstract: The subclass Cyrtophoria are a group of highly specialized ciliates, which inhabit soil, freshwater, brackish water, marine and other environments.

Please revise the use of "highly specialized ciliates" because later is reported that they inhabit everywhere (non-specialized). If there are specialized ciliates because must inhabit in a specific place. I know that you mean that the specialization is in the morphology and ultrastructure, but the sentence need to be reworded to be clear.

The use of "other environments" remains mysterious. Maybe you mean that they live in ice, so just cite the ice (you have no examples in China).

line 18: In almost a century of research history, there have been more than 50 publications on the diversity, taxonomy, phylogeny and ecology of cyrtophorian ciliates from China. These studies are revised here.

Better: We revised the 50 publications on the diversity, taxonomy, phylogeny and ecology of cyrtophorian ciliates in China since the earlier publication in 19xx.

If there is space in the abstract, you should report the 3 periods of the research history. Later, I will explain why I disagree about the periods.

line 22: 73 taxonomic hits.

line 304: 73 taxonomic hits (lowest taxonomic rank to species or genus)

In the abstract you should not introduce the ambiguous term "taxonomy hit" or at least please explain it.

line 23: Based on the morphological data from China and the existing molecular phylogeny analyses

You do not have to be based on the existing molecular phylogeny analyses. In a review you can use the available DNA sequences in GenBank in order to build your phylogenetic tree, and not the non-updated existing molecular phylogeny analyses.

Keywords: morphology; phylogeny; SSU rDNA

Where is the SSU rDNA phylogeny in this review?

line 41: (Figure 1, OK). I know that you refer to the oral kinety, but it is unclear to find (Figure 1, OK). Just report (see oral kineties (OK) in the Figure 1.

line 42; Figure 1, SZ) Similar for that.

The figure 1 explains the abbreviations, and the glossary at the end of the manuscript explains the terminology. Please decide if there is a table with all the abbreviations. The abbreviations should also appear in the glossary. Please report the abbreviations in the figure legends as you did in the figures 3, 4.

line 65: 180 nominal cyrtophorian ciliates

Please explain the term “nominal"

line 116, table 1. The publications should be in the last column, keeping the chorological order.

line 124:  figure 2 and line 128: the “standard” stage (C, since 1991).

I fully disagree with the periods. You can see a strong increase after year 2000. A period began after 2000. The "standard" stage corresponds to the "molecular period".

line 130: .2. Species List and Classification

If you are proposing a classification, it is mandatory that before you show a SSU rDNA gene phylogenetic tree to support your classification. You cannot based your classification mainly in Lynn 2008. There are many changes due to the molecular data since then. Please report an updated phylogenetic tree.

The figures 3 and 4 also need of a phylogenetic tree before.

line 191-201 Please use italic type for genus nad species names.

line 236: Chilodonella apouncinata nom. nov. 

It is missing the taxonomical authority. We do not know who the taxonomical authority of the new name is.

line 237: Please cite the figure of the type specimen.

Please report "non Chilodonella parauncinata Wang 1974” to remark that new names is due to a homonym.

You do not have to repeat the diagnosis, but you can cite the type locality and type material.

line 292: The figure 5 reflects a clear trend. About ¾ of China, on the western side, between Indian and Mongolia, and the northern region (Manchuria) remained unexplored. You must comment this geographical bias in the studies in the abstract, and to cite it as objectives in the last section named prospect.

line 296:  Molecular Phylogenetic Studies

This section should be before the classification and evolution

line 345 5. Proposed Phylogeny

The phylogenetic tree is missing before.

line 354: The figure 6 should be higher in order to have space to provide the genus name in each illustration.

line 408: 6.1. Life Styles

This section is poor documented. Please provide a table or fuse with a section with ecology.

line 443: 6.3. Abundance

In fact, there are no data. This is a part of a section on ecology.

line 460:   7. Prospect

As reported before, you must cite the important geographical bias (western China and Manchuria remain unexplored).

Ecologists have discussed for decades about the endemism or cosmopolitan distribution of microbes using the ciliates as example. Please comment about the possible endemisms in China.

Author Response

Reviewer 3

This manuscript is a review of the publications on the cyrtophorian ciliates from China. Certainly, the studies on ciliates have experienced an exponential increase in the last two decades in China, and this kind of reviews, with the schematic illustrations and glossary of the terminology, are also useful for the formation of researchers.

Authors: Many thanks for your critical comments and constructive suggestions that have greatly improved the quality of our manuscript.

However, this review has an important deficiency. There is no a phylogenetic tree (SSU rRNA gene) of this group of ciliates. This is needed because the authors proposed a classification of the cyrtophorian ciliates, and also scheme of the evolution. The reader needs to see an updated phylogenetic tree where to verify the coherence between the classification, evolution, and molecular data. This is the main reason to request a major revision. After that, I will recommend the publication of this review.

Authors: Many thanks for the constructive suggestion. In the original manuscript, we gave the classification and the proposed phylogeny based on our review of these molecular works as well as those morphological studies and our own knowledge. We totally agree with the reviewer that it would be much better if we reconstruct a phylogenetic tree using updated data, which will show more information to the readers. Hence, we reconstructed the phylogenetic trees using SSU rDNA sequences (Figure 5). In total 113 sequences were analyzed including 108 cyrtophorian, 2 chonotrich, and 3 suctorian (out groups) representatives. The sequence alignment and tree reconstruction methods are described in the Supplementary file. In brief, two algorithms, IQTREE and MrBayes, are employed to run the maximum likelihood analysis and Bayesian inference analysis, respectively. A paragraph was added in the section molecular phylogenetic studies (lines 371–379): “In the present work, we reconstructed a comprehensive phylogenetic tree inferred from updated SSU rDNA sequences (Figure 5). Sequence selection and alignment, and phylogenetic tree construction methods are described in Supplementary materials. The topology of the present phylogenetic tree is consistent with previous works [e.g., 13,40,58,76–80,85,87–89], and similar conclusions can be drawn: (1) The order Clamydodontida are monophyletic, while the order Dysteriida are non-monophyletic. (2) The subclass Chonotrichia are nested within Dysteriida, on the level of the family Hartmannulidae. (3) Kyaroikeidae display as a subfamily of Dysteriidae. (4) Gastronautidae represent a transitional family among Chilodonellidae, Chlamydodontidae, and Lynchellidae”

Minor comments:

line 17: Abstract: The subclass Cyrtophoria are a group of highly specialized ciliates, which inhabit soil, freshwater, brackish water, marine and other environments. Please revise the use of "highly specialized ciliates" because later is reported that they inhabit everywhere (non-specialized). If there are specialized ciliates because must inhabit in a specific place. I know that you mean that the specialization is in the morphology and ultrastructure, but the sentence need to be reworded to be clear.

Authors: Yes, we meant the specialization in the morphology. We have corrected it to “morphologically specialized”. Certainly, every group of ciliates is specialized in the morphology. We explain the morphological specialization of cyrtophorian ciliates in the introduction (lines 18, 43).

The use of "other environments" remains mysterious. Maybe you mean that they live in ice, so just cite the ice (you have no examples in China).
Authors: The sentence has been rewritten as: “The subclass Cyrtophoria are a group of morphologically specialized ciliates, which mainly inhabit soil, freshwater, brackish water, marine environments.” (Lines 18, 19).

line 18: In almost a century of research history, there have been more than 50 publications on the diversity, taxonomy, phylogeny and ecology of cyrtophorian ciliates from China. These studies are revised here.

Better: We revised the 50 publications on the diversity, taxonomy, phylogeny and ecology of cyrtophorian ciliates in China since the earlier publication in 19xx.

Authors: We accepted the suggestion and the sentence has been rewritten (lines 19–23).

If there is space in the abstract, you should report the 3 periods of the research history. Later, I will explain why I disagree about the periods.

Authors: One sentence reporting the three periods of the research history has been added in the abstract (lines 24, 25): “The research history can be divided into three periods: the early stage, the Tibet stage, and the standard stage.”

line 22: 73 taxonomic hits.

line 304: 73 taxonomic hits (lowest taxonomic rank to species or genus)

In the abstract you should not introduce the ambiguous term "taxonomy hit" or at least please explain it.

Authors: Agree! We added brackets “(lowest taxonomic rank to species or genus)” to explain the term taxonomic hits (line 28).

line 23: Based on the morphological data from China and the existing molecular phylogeny analyses. You do not have to be based on the existing molecular phylogeny analyses. In a review you can use the available DNA sequences in GenBank in order to build your phylogenetic tree, and not the non-updated existing molecular phylogeny analyses.

Authors: We agree with the reviewer and the word “existing” has been deleted (line 30).

Keywords: morphology; phylogeny; SSU rDNA

Where is the SSU rDNA phylogeny in this review?

Authors: These contents were reviewed in the original manuscript (323–370). In addition, we added our own molecular phylogenetic analyses based on SSU rDNA sequences (in the section Molecular Phylogenetic Studies.

line 41: (Figure 1, OK). I know that you refer to the oral kinety, but it is unclear to find (Figure 1, OK). Just report (see oral kineties (OK) in the Figure 1.

line 42; Figure 1, SZ) Similar for that.

Authors: The expression the reviewer suggested is much better. We made corrections accordingly (lines 48, 49, 51). Thanks!

The figure 1 explains the abbreviations, and the glossary at the end of the manuscript explains the terminology. Please decide if there is a table with all the abbreviations. The abbreviations should also appear in the glossary. Please report the abbreviations in the figure legends as you did in the figures 3, 4.

Authors: Thanks for the advice. We decided to move the abbreviations in the Figure 1 to the figure legends (lines 64, 65), as in Figures 3, 4. We did not give a table to explain the abbreviations, but instead, we added abbreviations in the glossary (lines 851–887).

line 65: 180 nominal cyrtophorian ciliates

Please explain the term “nominal"

Authors: There are about 180 cyrtophorian species reported, but we did not check if all these are valid. Thus, we use “nominal” to differ them from valid species. But now, obviously, it causes misunderstanding. So, we decided to delete the word “nominal” (line 77).

line 116, table 1. The publications should be in the last column, keeping the chorological order.

Authors: Many thanks for the suggestion. We wanted to present the list in the order of year and authors, which will show the sequence of the publications. So, we decided to keep the publications in the first column in chorological order. Many species were reported in different publications. If the publications are in the last column, then the species names will be presented in the first column; the current non-alphabet order of the species names will be confusing.

line 124:  figure 2 and line 128: the “standard” stage (C, since 1991).

I fully disagree with the periods. You can see a strong increase after year 2000. A period began after 2000. The "standard" stage corresponds to the "molecular period".

Authors: The comment sounds reasonable and we accepted the suggestion. Corrections were made accordingly (lines 120–128).

line 130: .2. Species List and Classification

If you are proposing a classification, it is mandatory that before you show a SSU rDNA gene phylogenetic tree to support your classification. You cannot based your classification mainly in Lynn 2008. There are many changes due to the molecular data since then. Please report an updated phylogenetic tree. The figures 3 and 4 also need of a phylogenetic tree before.

Authors: We reworded this part because we did not mean to revise the classification, but instead, we only list the genus classification by Lynn (2008) and later modifications (references in Table 2). We presented figures 3 and 4 as morphological keys, but not exactly the phylogenetic patterns. A phylogenetic tree (Figure 5) has been added in the section Molecular phylogenetic studies.

line 191-201 Please use italic type for genus nad species names.

Authors: Corrected (lines 208–217).

line 236: Chilodonella apouncinata nom. nov. 

It is missing the taxonomical authority. We do not know who the taxonomical authority of the new name is.

Authors: “nov.” indicates that the name is newly proposed in our paper; thus, we (the current authors) are the authority of the new name.

line 237: Please cite the figure of the type specimen.

Authors: Done (line 262).

Please report "non Chilodonella parauncinata Wang 1974” to remark that new names is due to a homonym.

Authors: Corrected, thanks! (Lines 255, 256).

You do not have to repeat the diagnosis, but you can cite the type locality and type material.

Authors: Diagnosis deleted (line 261).

line 292: The figure 5 reflects a clear trend. About ¾ of China, on the western side, between Indian and Mongolia, and the northern region (Manchuria) remained unexplored. You must comment this geographical bias in the studies in the abstract, and to cite it as objectives in the last section named prospect.

Authors: The geographical bias was already highlighted in the original manuscript in the section 3. Sites, Habitats, and Distribution (lines 488–450): “However, apart from the very early stage, respective investigations have been mostly carried out along the coastal areas (Figure 7). While other habitats were only sporadically studied.” And this was also mentioned in the last section Prospect, and we made some corrections based on your suggestion: (lines 551–555): “Thus, more investigations of the fauna need to be conducted on a large regional scale with different habitats. In contrast to massive investigations from marine and brackish water habitats, sampling from freshwater and soil as well as extreme environments like hypersaline and cold regions are urgently needed.”

line 296:  Molecular Phylogenetic Studies

This section should be before the classification and evolution

Authors: Thanks for the suggestion. The classification was summarized from the morphological taxonomy works, and we have no further modification. So, it would be better to put this part in the section of morphological studies. As mentioned above, an updated phylogenetic tree was added in the R1, and the section molecular phylogenetic studies was before the section evolution (proposed phylogeny).

line 345 5. Proposed Phylogeny

The phylogenetic tree is missing before.

Authors: A phylogenetic tree inferred from SSU rDNA sequences has been provided.

line 354: The figure 6 should be higher in order to have space to provide the genus name in each illustration.

Authors: Thanks for the advice. We tried this before but found the space was too crowded with the genus names in the illustrations. So, we decided to show the genus name on the right.

line 408: 6.1. Life Styles

This section is poor documented. Please provide a table or fuse with a section with ecology.
Authors: This was already a part in the section ecology in the original manuscript.

line 443: 6.3. Abundance

In fact, there are no data. This is a part of a section on ecology.

Auhtors: This was already a part of section on ecology in the original manuscript.

line 460: 7. Prospect

As reported before, you must cite the important geographical bias (western China and Manchuria remain unexplored).

Authors: Done, thanks.

Ecologists have discussed for decades about the endemism or cosmopolitan distribution of microbes using the ciliates as example. Please comment about the possible endemisms in China.

Authors: Thanks for the advice. We agree with the reviewer that the distribution pattern is one of the key issues the ecologists (especially those in the field of microbiology) want to solve. While, this has been already presented in the original manuscript in the section Sites, Habitats, and Distribution (now lines 468–484 in R1): “Many studies on pelagic and soil habitats indicate that the dispersal and distribution of ciliates follow the moderate endemicity model (e.g., [92]). Apparently, this also holds true for cyrtophorian ciliates. Currently, more than one third of cyrtophorians discovered in China are new and possibly endemic, while the rest can be found on other continents as well. The only attempt to analyze the biogeographic distribution patterns of cyrtophorian ciliates was briefly conducted on a well-studied genus, Chlamydodon [12]. This work summarized the historical studies on morphology worldwide, showed the global distribution of Chlamydodon species, and indicated possible cosmopolitan and endemic species. But as demonstrated by the authors, the analyses were very limited, mainly because of the scattered studies on this genus. Therefore, in order to obtain more detailed data (at population level) with a higher geographical resolution, extensive sampling must be carried out in different habitats at a larger scale. This limit also applies to the studies on cyrtophorian ciliates in China. For instance, a cosmopolitan species, Chlamydodon mnemosyne, has only been formally reported once in publication from China [61], but sampled and recorded in master and PhD theses for several times (by personal communication). This issue prevents a thorough diversity study and also hinders intra- and interspecific comparisons, for both, morphological and molecular aspects. Therefore, in order to obtain more detailed data (at population level) with a higher geographical resolution, extensive sampling must be carried out in different habitats at a larger scale.”

Round 2

Reviewer 3 Report

The authors have included almost all the suggestions of this Reviewer, especially a molecular phylogeny tree, and some parts of the text have been reorganized. The period from 1991 to the present have been modified as suggested, now from 2000 to the present. However, this period from year 2000 is the molecular period. The use of standard period is not informative because –standard- has not a clear meaning. This review is almost ready for acceptance, after the authors will consider a few minor suggestions.

Abstract

line 19: We revised more than 50 publications

line 98: For this section, we collected and reviewed 50 publications (Table 1)

Please decide if you reviewed 50 or more than 50 publications.

line 20: on the diversity, taxonomy

line 88 aspects of diversity, taxonomy,

line 546 diversity, taxonomy,

What is the differences between studies on diversity or taxonomy?

line 25: and the standard stage

As reported before, the use of standard is not informative, the current period is the molecular one.

line 66: bacteria (including cyanobacteria) and/or microalgae [10–13]

Cyanobacteria are microalgae, then use "and/or eukaryotic microalgae".

line 73: freshwater fish agriculture tanks

the tanks are not for agriculture, the tanks are for aquaculture.

line 77: To date, about more than 180 cyrtophorian ciliates are known worldwide

180 cyrtophorian ciliate species/taxa?

line 86: SSU rDNA sequences, and the rest of the text.

Contrary to other tentative comments by other Reviewers, the correct terminology is SSU rRNA gene sequences. The correction in line 360 is unnecessary.

line 124: most species (populations)

Species or populations?

line 231, 235: The arrows and colored circles indicate dichotomic characteristics

dichotomic characteristics or diagnostic characters?

line 254: Chilodonella apouncinata nom. nov.

As requested in the previous review, please provide the taxonomical authorities for the new species name.

line 335: For a better visualization of the phylogenetic tree, please provide the tree file in eps format.

line 401. Figure 6. The list of genera should have a larger type.

line 408. But nothing However, nothing

line 476: But as stated However, as stated

line 480: master Master

line 492: 20ncinate

line 497: Figure 7. The yellow squares are not needed.

Author Response

Dear reviewer,

Please find enclosed our revised manuscript We thank again the anonymous reviewers, especially the reviewer 3, for providing extremely thoughtful, considerate, and constructive comments and suggestions. Without exception, all comments were helpful to improve the manuscript and we hope the reviewer 3 will be satisfied with the revised manuscript. We are very grateful for this review process and want to express our gratitude in the

Acknowledgements: We would like to thank the anonymous reviewers for their constructive comments, which helped to improve the manuscript.

Reviewer 3:

The authors have included almost all the suggestions of this Reviewer, especially a molecular phylogeny tree, and some parts of the text have been reorganized. The period from 1991 to the present have been modified as suggested, now from 2000 to the present. However, this period from year 2000 is the molecular period. The use of standard period is not informative because –standard- has not a clear meaning. This review is almost ready for acceptance, after the authors will consider a few minor suggestions.

Authors: Thanks for your comments. We have used molecular stage instead of standard period, as suggested.

Abstract

line 19: We revised more than 50 publications

line 98: For this section, we collected and reviewed 50 publications (Table 1)

Please decide if you reviewed 50 or more than 50 publications.

Authors: Thank you for the notice of the different publication numbers. For the section 2. Morphological Taxonomy, we revised exactly 50 publications, as listed in Table 1. These publications are dealing with morphology of cyrtophorian ciliates. However, apart from these 50 publications, we also revised other papers on molecular phylogenetic research, e.g., Gao et al. (2012), Chen et al. (2016), and Wang et al. (2017), which were not revised in the section Morphological Taxonomy. Thus, we wrote we revised more than 50 publications in the abstract.

line 20: on the diversity, taxonomy

line 88 aspects of diversity, taxonomy,

line 546 diversity, taxonomy,

What is the differences between studies on diversity or taxonomy?

Authors: Thanks for the question. We thought the taxonomy work contributed to the “diversity” study, so we used “diversity and taxonomy”. However, after careful consideration, we realized that apart from the aspect of richness, we did not provide any other information on diversity, especially the quantity. Thus, we decided to delete the term diversity.

line 25: and the standard stage

As reported before, the use of standard is not informative, the current period is the molecular one.

Authors: Corrected.

line 66: bacteria (including cyanobacteria) and/or microalgae [10–13]

Cyanobacteria are microalgae, then use "and/or eukaryotic microalgae".

Authors: Corrected (lines 60, 449, and 450).

line 73: freshwater fish agriculture tanks

the tanks are not for agriculture, the tanks are for aquaculture.

Authors: Corrected (line 67).

line 77: To date, about more than 180 cyrtophorian ciliates are known worldwide

180 cyrtophorian ciliate species/taxa?

Authors: Species. Corrected (line 70).

line 86: SSU rDNA sequences, and the rest of the text.

Contrary to other tentative comments by other Reviewers, the correct terminology is SSU rRNA gene sequences. The correction in line 360 is unnecessary.

Authors: Thanks for your opinion. According to the Wikipedia (https://en.m.wikipedia.org/wiki/Ribosomal_DNA): Ribosomal DNA (rDNA) is a DNA sequence that codes for ribosomal RNA. So, as we understand, rDNA equals to rRNA gene, and both are correct. In order to keep the consistency, we used rDNA throughout the manuscript.

line 124: most species (populations)

Species or populations?

Authors: “Populations” was deleted.

line 231, 235: The arrows and colored circles indicate dichotomic characteristics

dichotomic characteristics or diagnostic characters?

Authors: We think dichotomic is proper. Those characters are used to efficiently distinguish the two branches, which the readers can use to identify a genus quickly. That’s why we termed them “dichotomic”. They are better, but not necessarily, to be diagnostic characters.

line 254: Chilodonella apouncinata nom. nov.

As requested in the previous review, please provide the taxonomical authorities for the new species name.

Authors: As replied in the previous version, “nov.” indicates that the name is newly proposed in our paper; thus, we (the current authors) are the authorities of the new species name. This a common practice in ciliate taxonomy.

line 335: For a better visualization of the phylogenetic tree, please provide the tree file in eps format.

Authors: The figure resolution in the revised manuscript was low, although we provided high resolution version in the submission system (.jpg format, 5.6 Mb). We tried the eps format as you requested, but we found its size was over 100 Mb. Too big for the submission system. Instead, we can provide the figure here, for your reference. Also, we made it bigger in the R2.

line 401. Figure 6. The list of genera should have a larger type.

Authors: Thanks for the suggestion. Now, the Figure 6 has been modified to have a larger genus list.

line 408. But nothing However, nothing

line 476: But as stated However, as stated

line 480: master Master

Authors: Suggestions accepted, thanks!

line 492: 20ncinate

Authors: Corrected to uncinata.

line 497: Figure 7. The yellow squares are not needed.

Authors: The yellow squares were removed. Thanks for the observation!